# Carbon-Based Enzyme Mimetics for Electrochemical Biosensing

**DOI:** 10.3390/mi14091746

**Published:** 2023-09-07

**Authors:** Esther Sánchez-Tirado, Paloma Yáñez-Sedeño, José Manuel Pingarrón

**Affiliations:** Department of Analytical Chemistry, Faculty of Chemistry, University Complutense of Madrid, 28040 Madrid, Spain; esther.sanchez@ucm.es (E.S.-T.); pingarro@quim.ucm.es (J.M.P.)

**Keywords:** carbon nanozyme, artificial enzyme, enzyme mimicking, electrochemical biosensor

## Abstract

Natural enzymes are used as special reagents for the preparation of electrochemical (bio)sensors due to their ability to catalyze processes, improving the selectivity of detection. However, some drawbacks, such as denaturation in harsh experimental conditions and their rapid de- gradation, as well as the high cost and difficulties in recycling them, restrict their practical applications. Nowadays, the use of artificial enzymes, mostly based on nanomaterials, mimicking the functions of natural products, has been growing. These so-called nanozymes present several advantages over natural enzymes, such as enhanced stability, low cost, easy production, and rapid activity. These outstanding features are responsible for their widespread use in areas such as catalysis, energy, imaging, sensing, or biomedicine. These materials can be divided into two main groups: metal and carbon-based nanozymes. The latter provides additional advantages compared to metal nanozymes, i.e., stable and tuneable activity and good biocompatibility, mimicking enzyme activities such as those of peroxidase, catalase, oxidase, superoxide dismutase, nuclease, or phosphatase. In this review article, we have focused on the use of carbon-based nanozymes for the preparation of electrochemical (bio)sensors. The main features of the most recent applications have been revised and illustrated with examples selected from the literature over the last four years (since 2020).

## 1. Introduction

In 1970, Breslow and Overman [1] introduced the concept of artificial enzymes as a combination of a metal catalytic group and a hydrophobic binding cavity. More recently, this concept was extended to a variety of nanomaterials, and the name nanozyme was adopted. Among the nanomaterials described as catalysts with different enzyme activities, those functioning as peroxidase, catalase, oxidase, superoxide dismutase, nuclease, or phosphatase should be highlighted. Nowadays, these materials can be divided into two main groups: metal and carbon-based nanozymes [2]. Several advantages of nanozymes over natural enzymes, such as enhanced stability, low cost, facile production, and rapid activity, can be mentioned. These outstanding features are responsible for their widespread use in areas such as catalysis, energy, imaging, sensing, or biomedicine. Various review articles were recently published on the characteristics and applications of nanozymes. For instance, the enzyme-mimicking activities of different nanomaterials and their application to the fabrication of biosensors [3,4] and the detection of clinical biomarkers [5], as well as the applications of nanozymes in environmental monitoring [6], were reviewed. The recent progress of nanozymes in electrochemical sensing [7,8] and biosensing [9] was also reviewed.

Regarding carbon-based nanozymes, their catalytic activity has been extensively reported [10], and several examples of applications as metal-free catalysts for a variety of (bio)chemical reactions have been described [11]. Carbon nanomaterials, including fullerenes, carbon nanotubes, graphene, graphene oxide, carbon dots, graphene quantum dots, carbon nitrides, and their hybrids [12], have abundant active sites and possess special electronic and geometric properties. Furthermore, they are characterized by good stability and biological safety, while their intrinsic enzyme activity allows them to mimic natural oxidase, peroxidase, superoxide dismutase, or catalase enzymes [7]. These nanomaterials provide additional advantages compared to metal-based artificial enzymes, such as higher stability, tuneable activity, and good biocompatibility. Progress on the use of peroxidase-like carbon-based nanozymes [13] and their biomedical applications [14] have been reviewed. More recently, Sun et al. [12] revised the design, catalytic mechanism, and bio-application of carbon nanozymes.

In this review article, we have focused on the use of carbon-based nanozymes for the preparation of electrochemical (bio)sensors. The main characteristics and advantages offered by the most recent applications have been illustrated with examples selected from the literature in the last four years (since 2020). The highlighted approaches have been ordered according to the type of carbon nanomaterial and their use as electrode modifiers or nanocarrier tags for detection, as well as the class of (bio)sensing platform. To provide complete information, the tables summarize the rationale and main features of relevant (bio)sensing devices involving carbon nanozymes applied mainly to clinical, food, and environmental monitoring.

## 2. Nanozymes Involving Carbon Nanotubes

As stated above, carbon nanotubes, graphene, and graphene oxide are some of the carbon nanomaterials whose enzymatic activity has been known for years and that have found application in the construction of electrochemical (bio)sensors. Although fullerenes have also been shown to possess enzyme-like behavior in the form of superoxide dismutase (SOD) [15], no applications based on this property have been found in the (bio)electroanalytical field. It should be noted that the enzymatic activity of fullerenes has been attributed to the existence of surface electron defect regions that promote the adsorption of O_2_**^·^**^−^ [16], being also dependent on the number of carboxyl groups on carboxyfullerenes [17].

Regarding carbon nanotubes, some recent and interesting applications should be noted. Song et al. [18] described the intrinsic peroxidase-like activity of single-walled carbon nanotubes catalyzing the reaction of the peroxidase substrate 3,3,5,5-tetramethyl-benzydine (TMB) in the presence of hydrogen peroxide. In addition, the effect of the different oxygenated groups on the enzymatic activity of this nanomaterial was investigated [19]. Although the advantages of these nanomaterials are significant, the main disadvantage is the relatively low enzyme activity, which has led to the design of several strategies to enhance their catalytic effects. Among them, doping with heteroatoms should be highlighted due to its effectiveness. In various configurations, carbon nanomaterials act as supports for metal nanoparticles that enhance their mimicking activity. As an example, Co nanoparticle-decorated bamboo-like N-doped carbon nanotubes (Co-bNCNTs) exhibited excellent oxidase activity, prevented self-aggregation, and exposed more active sites [20]. Moreover, the synergistic effect between CoNPs and bNCNTs could also contribute to enhancing the catalytic activity. The Co-bNCNTs/GCE sensor was applied to the determination of dopamine (DA), showing a wide detection range of 0.5–150 μM and a low detection limit of 0.0342 μM. The Michaelis–Menten constant, K_m_, was also determined from the Lineweaver–Burk equation, which provided a value of 87.48 μM for DA.

More recently, an oxidase-like nanozyme was prepared from boron–nitrogen-doped CNTs with encapsulated iron nanoparticles, Fe/BNCNTs [21]. Using this nanomaterial, an electrochemical sensor was fabricated for the simultaneous determination of DA and uric acid (UA) with a peak potential separation of 139 mV. The calibration curves ranged from 1 to 630 μM (DA) and 0.5 to 2065 μM (UA), with detection limits of 0.8 and 0.28 μM, respectively. In addition, the proposed sensor was successfully applied to the detection of both targets in human serum. Ruthenium nanoparticles have also been utilized to prepare carbon nanotube hybrids for catalytic applications. For instance, MWCNTs/RuNPs produced a synergic effect on the non-enzymatic catalytic reduction of H_2_O_2,_ allowing quantification in a wide linear range from 0.5 μM to 1.75 mM. This good behavior contributed toward developing a bienzymatic glucose biosensor in the presence of avidin and biotinylated glucose oxidase [22].

Artificial nanozymes involving carbon nanotubes combined with other nanomaterials have been extensively reported. Lin and Lin prepared a CNT-based nanozyme by cyclic voltammetric functionalization of CNTs with 2,20-azino-bis(3-ethyl-benzothiazoline-6-sulfonic acid) (ABTS). The corresponding sensor was applied to the determination of human serum albumin (HSA) in neat urine [23]. The kinetics of adsorption and the electrochemical oxidation of HSA at the modified electrodes were investigated, and a sensitive method was developed for the point-of-care diagnosis of microalbuminuria. A porous MoS_2_/MWCNT nanohybrid network with oxidase-like behavior was used to develop an electrochemical nanozyme sensor coupled with machine learning for the DPV detection of carbendazim (CBZ) in tea and rice [24]. Good electrocatalytic capacity and a wide linear range between 0.04 and 100 μM CBZ with an LOD value of 7.4 nM were obtained. More recently, a nanozyme with 2D/1D heterostructure was fabricated by the in situ growth of MoS_2_ nanosheets onto single-walled carbon nanotubes (SWCNTs). The so-obtained SWCNTs@MoS_2_ nanozyme exhibited greatly improved peroxidase-like activity due to the 2D/1D interfacial coupling, which might provide more active sites enhancing charge transferring during the catalytic reactions [25]. MoS_2_/MWCNTs nanostructures were prepared by our group and used as nanocarrier tags with immobilized detection antibodies for the construction of a dual immunosensor for the simultaneous determination of two cytokines, BAFF (B cell activation factor) and APRIL (a proliferation-induced signal) [26]. The peroxidase-like catalytic activity of the MoS_2_-based nanomaterials was positively evaluated by testing the oxidation of TMB in the presence of H_2_O_2_. However, a weak enzymatic activity was observed, and so, in order to obtain an electrochemical response high enough to reach the required sensitivity, HRP was also incorporated into the MoS_2_/MWCNTs conjugate. Table 1 summarizes the characteristics of the selected methods commented above as well as of other recently reported applications of carbon nanotube-based nanozymes [20,21,23,24,26].

## 3. Nanozymes Involving Graphene-Based Nanomaterials

Graphene oxide (GO) and reduced graphene oxide (rGO), with the two-dimensional structure of graphene, also play key roles in the nanozyme field. They have abundant surface defects and various oxygen-containing functional groups, including epoxy, hydroxyl, ether, endoperoxide, carbonyl, carboxyl, and ester, which have been reported to display peroxidase mimicking activities. Song et al. stated that carboxyl-modified GO showed intrinsic peroxidase-like activity and applied it to the colorimetric detection of glucose [27]. GO and rGO can activate H_2_O_2_, generating hydroxyl radicals with higher oxidizabilities and performing POD-mimicking activities [28]. N-doping in carbon nanomaterials also selectively enhances the peroxidase-like activities of rGO, most likely due to the adjustment of the charge density and the increase in active sites [29]. As an example, Hu et al. [30] compared the kinetic parameters of the enzyme-like behavior for rGO and N-doped rGO, obtaining Michaelis–Menten constants (K_m_) of 0.5456 mM and 0.1115 mM for H_2_O_2_, respectively, the lower value indicating a higher affinity for the substrate.

The origin of the peroxidase-mimicking activity of graphene-based nanomaterials such as GO and rGO is not well known [28]. Some studies indicate that, among the various oxygen moieties, the carbonyl groups are the active centers. Several graphene-based hybrids have been developed for their use as nanozymes and applied to the construction of electrochemical sensors and biosensors. However, the enzymatic activity has not been demonstrated in all cases. Carboxyl-modified graphene oxide (GO-COOH) has been found to exhibit intrinsic peroxidase-like activity when catalyzing the reaction of the peroxidase substrate TMB in the presence of H_2_O_2_ [5]. Song et al. [27] determined low Michaelis–Menten constants, K_m_, of 0.0237 mM and 3.99 mM for GO-COOH using TMB or H_2_O_2_ as the respective substrates. Accordingly, some electrochemical biosensors involving GO-COOH and GO-COOH hybrids have been developed. For instance, Dilmac and Guler [31] synthesized a nanocomposite by deposition of AuNPs onto GO-COOH, which was applied to the enzyme-free electrochemical oxidation of glucose in an alkaline medium. The sensor exhibited a wide linear range of 0.02–4.58 mM with an LOD value of 6 μM. An artificial peroxidase-like hybrid involving hemin-rGO (H-rGO) and PdNPs was prepared and used for signal amplification in the construction of an electrochemical biosensor for the determination of glypican 3 (GPC3), an emerging biomarker of hepatocellular carcinoma (HCC) [32]. As Figure 1 shows, the H-rGO-PdNPs nanozyme was conjugated to a specific aptamer (GPC3-Apt) to establish a sandwich-type configuration. Good peroxidase activity was demonstrated by enhancing H_2_O_2_ to reduce silver ions on the electrode surface. The amount of deposited Ag was proportional to the concentration of GPC3 with an LOD value of 3.30 ng mL^−1^ GPC3. 

Metallic oxides, chalcogenides, and other types of nanomaterials have also been dispersed and stabilized onto graphene derivatives to prepare nanozymes with a variety of catalytic activities. For instance, a polydopamine (PDA) surface modification-assisted pyrolysis strategy was employed to prepare CoO nanoparticle/N-doped carbon sheets/reduced graphene oxide composites (CoONPs/N-CS/rGO). In this strategy, PDA, with abundant amine and imine moieties, served as an eco-friendly N-contained precursor and a connecting agent between the GO matrix and CoONPs [33]. Due to the high conductivity of N-CS/rGO, the synergistic catalysis between CoONPs nanoparticles and N-CS/rGO, and the intrinsic oxidase-like properties, the CoONPs/N-CS-rGO nanocomposite was used for signal amplification in the simultaneous analysis of DA and UA. The apparent Michaelis−Menten constants determined for these substrates were, respectively, K_m_ = 35.38 μM and 51.17 μM. These relatively low values reveal its specific recognition ability toward the target molecules.

Laccase is a copper ion-containing oxidase-type enzyme that offers remarkable selectivity towards phenolic and polyphenolic substrates. Gugoasa et al. [34] prepared a AuNP/rGO hybrid laccase-like nanozyme and compared its activity to detect catechol as the target phenolic compound with that of the natural enzyme. The Michaelis–Menten constant was found to be 1.87 mM for the nanozyme deposited onto a screen-printed carbon electrode, a value slightly lower than that obtained for the native laccase, 5 mM. This result implied a stronger interaction between the substrate and AuNPs/rGO. More recently, copper nanocubes (CuNCs) were electrodeposited onto few- or multilayer graphene to obtain CuNCs-Gr/SPCE nanozyme platforms exhibiting laccase or tyrosinase-like oxidase activity. The developed nanozyme platform was used for the determination of dopamine in biological samples. The linear range was between 1 nM and 0.1 mM, and the LOD value was 0.33 nM [35]. 

The role of the graphene derivatives forming hybrid or artificial composite nanozymes by combination with other materials in the enzyme activity of the set is not sufficiently known. These carbon nanomaterials are good conductors and act as stabilizing agents preventing agglomeration, but exhibit weak enzyme-like activity, and thus, their contribution to the enzymatic catalysis provided by the whole nanomaterial needs to be investigated to address this knowledge towards obtaining more efficient artificial enzymes. In this sense, it is worth highlighting the work of Zhang et al. [36], who investigated the effect of rGO on the enzymatic activity of Prussian Blue nanoparticles (PBNPs), finding that the rGO structuring interface altered the properties of the nanomaterial for nanozyme-based sensing approaches. For example, electrochemical sensors using a GCE/(PB/rGO)/Nafion electrode exhibited a lower sensitivity but a wider linear range up to higher concentrations of hydrogen peroxide at pH 7.4 when compared with GCE/PB NPs/Nafion.

Table 2 summarizes the characteristics of the selected methods commented above, as well as of other recently reported applications of graphene-based nanozymes [31,32,33,34,35,36,37,38,39,40].

## 4. Nanozymes Involving Graphene Quantum Dots and Carbon Dots

The oxidase- and peroxidase-mimicking activity of graphene quantum dots (GQDs) and carbon dots (CDs) has been explored for analytical (bio)sensing using electrochemical transduction and other detection techniques. These nanozymes can replace natural pero-xidase-based systems such as HRP (horseradish peroxidase)/H_2_O_2_, which requires labeling to a receptor for target determination, thus allowing label-free detection [41]. Both particles are defined as zero-dimensional nanomaterials showing low toxicity and biocompatibility, but unlike CDs, GQDs possess a graphene structure, which confers them the unusual properties of graphene. As in the previous section, with the aim to illustrate the importance of these nanomaterials in the construction of electrochemical (bio)sensors, several selected examples reported in the literature during recent years are summarized in Table 3 [42,43,44,45,46,47,48,49,50,51,52,53]. Although the selected examples involved electrochemical transduction, it should be noted that the number of applications using colorimetric detection where the enzymatic activity of the nanomaterial is detected by the oxidation and color change of TMB is much larger [54,55]. On the other hand, most of the reported electrochemical me-thods involve the use of hybridized or doped GQDs and CDs with other nanomaterials, mainly metal nanoparticles or oxides, that modulate and greatly enhance enzymatic activity.

In an interesting article, a nanocomposite of manganese dioxide and GQDs was used as a dual-functional sensing platform mimicking oxidase activity for the electrochemical and colorimetric detection of catechol (CC) and DA. The kinetic parameters were determined using TMB as the target, yielding a low K_m_ value of 0.11 mM. A linear relationship between the current signal and the concentration of DA and CC was obtained by DPV over the 0.5–100 μM and 5–150 μM ranges, with limits of detection of 0.05 µM and 0.09 µM, respectively [42]. The authors concluded that the electrochemical technique allowed a better sensitivity toward both analytes, with wider linear ranges, lower limits of detection, and reliability for the detection of phenolic compounds in real environmental samples. As seen in Table 2, various configurations involving AuNPs have been recently developed. AuNP/GQD nanozyme-modified electrodes contribute to the high catalytic activity and sensitivity provided by both nanocomposite elements, as is the case of the electrochemical sensor developed for the determination of quercetin by square-wave voltammetry (SWV), which provides a wide linear range of 0.1 nmol L^−1^ to 1.0 × 10^−3^ mol L^−1^ and a limit of detection of 0.033 nmol L^−1^. The method was applied to the analysis of spiked human plasma with the advantage of requiring only a droplet of the sample [43]. More recently, GQDs synthesized by a green hydrothermal method using citric acid were mo-dified with AuNPs and 1-amino-2-naphthol-4-sulfonic acid (ANSA) and utilized for the preparation of an electrochemical sensor for the determination of methotrexate (MTX), an anti-cancer drug [44]. The catalytic capability and synergistic effect of the ANSA/AuNP/GQD nanocomposite deposited onto a glassy carbon electrode made possible the determination of MTX over the 0.1–100 μmol L^−1^ linear range with an LOD value of 0.03 μmol L^−1^.

Metal−organic frameworks (MOFs) are characterized by a large specific surface area, a high number of well-dispersed active sites, and controllable functionalities and porosity [56]. These properties make MOFs to be used as efficient supports for catalysts immobilization and enzymatic regulation [57]. As an example, a nanocomposite of nitrogen-doped GQDs and AuNPs (AuNPs/N-GQDs) with high peroxidase mimicking activity was anchored on the surface of a polyethyleneimine-functionalized MOF (P-MOF) which provided a large surface area for AuNP loading and facilitated the contact between the active sites and the electrode surface. A typical Michaelis–Menten behavior using TMB or H_2_O_2_ was found for the AuNP/N-GQD nanozymes, with apparent K_m_ constants of 0.056 and 0.469 mmol L^−1^, respectively, suggesting strong affinity toward the target molecules [45]. Interestingly, the high peroxidase-like activity and large specific surface area of AuNPs/N-GQDs-P-MOF allowed the preparation of a nanocarrier for glucose oxidase (GOx) by immobilizing the enzyme to achieve a cascade reaction amplification for glucose detection (Figure 2). The intermediate by-product (H_2_O_2_) generated from the enzymatic glucose oxi-dation can be decomposed into H_2_O and oxygen in the presence of AuNP/N-GQD nanozyme. Then, the oxygen can be re-used in the enzymatic glucose oxidation to increase the reaction rate. The amplified amperometric glucose biosensor showed an LOD value of 0.7 μM [45].

CDs are excellent nanomaterials for the preparation of hybrids or composites with enzyme-mimicking activity due to their easy surface modification and doping with hete-roatoms [58,59]. The resulting nanozymes have been shown to exhibit oxidase, catalase, and superoxide dismutase, among other catalytic functions. The recent progress in CDs-based nanozymes for chemosensing and biomedical applications was reviewed by He et al. [60]. A variety of designs involving different combined nanomaterials to enhance the enzyme-like activity, as well as the electrocatalytic effects and sensitivity, have been reported in recent years for their use in the preparation of electrochemical (bio)sensors. As an interesting application, a biosensor for the detection of virulence outer membrane protein A (ompA) gene of Cronobacter sakazakii (*C. sakazakii*) was developed based on the peroxidase-mimicking activity of boron-doped carbon dots–Au nanoparticles (B-CDs-AuNPs) and a signal amplification strategy of exonuclease III (Exo III)-assisted target-recycling. The electrochemical response was obtained from the reduction of H_2_O_2_ catalyzed by B-CDs-AuNP nanozyme and provided a high sensitivity with a broad linear range from 0.001 to 10 pmol L^−1^ and a limit of detection as low as 0.01 fmol L^−1^. Interestingly, the biosensor was applied to the direct detection of extracted DNA from *C. sakazakii*, obtaining a linear semilogarithmic calibration plot between 7.8 and 7.8 × 106 CFU mL^−1^ and an LOD value of 2.6 CFU mL^−1^. The developed method was applied to the determination of ompA gene segments in contaminated infant formula [48].

A photoelectrochemical biosensor involving multifunctional Au@Ag@CDs peroxidase nanoenzymes coupled with Ti_2_C/Ag_2_S composites was developed for the detection of microRNA [49]. As Figure 3 shows, after immobilization of a hairpin subsidiary probe (S1) onto the Ti_2_C/Ag_2_S surface, the target microRNA-associated bipedal DNA performed the opening of S1 to form triple helix molecules. Then, the Au@Ag@CDs were introduced into the biosensing platform through target-induced triple helix molecules. The Au@Ag@CDs nanocomposite acted as a multifunctional signal amplifier, which exhibited a synergistic catalytic effect as a peroxidase nanoenzyme. Based on the plasmonic Au@Ag@CDs nanoenzymes coupled with Ti_2_C/Ag_2_S composites, microRNA-155 was detected within a wide 1–10,000 fmol L^−1^ linear range with a low limit of detection of 0.83 fmol L^−1^.

Li et al. [50] proposed a double catalytic amplification strategy combining Fe_3_O_4_/CDs nanozymes and Ag-MOFs for the preparation of an electrochemical microfluidic paper-based chip to detect parathion-methyl (PM). In addition to the enzymatic catalysis coupled with the amplification of the response promoted by both components, the authors designed a molecularly imprinted polymer (MIP) using PM as a template molecule for improving selectivity. The sample introduced into the reaction zone was captured by the MIP, and a reduction current at −0.53 V was generated from the target in the presence of hydrogen peroxide, which was detected at a gold electrode. A mechanism for the double amplification effect was proposed in which the catalyzed oxidation of H_2_O_2_ produces se-veral electrons and protons whose presence accelerated the reduction reaction of PM to attain a larger current and provide a detection limit of around 10 pmol L^−1^.

Table 3 shows that other nanomaterials, such as metal oxides [51,52] and sulfides [53], combined with carbon dots have been proposed as nanozymes for the construction of electrochemical (bio)sensors. As an example, Honarasa et al. [52] prepared a triple nanozyme configuration involving Fe_3_O_4_, CeO_2_, and CDs exhibiting horseradish peroxidase-like catalytic activity with a Michaelis–Menten constant as low as 67.2 mM using H_2_O_2_ as substrate. On the other hand, ultrasmall Co_9_S_8_ nanocrystals stabilized on CDs have been synthesized, resulting in nanocomposites with a rich pore structure and outstanding bifunctional performances to mimic the catalytic activity of peroxidase and oxidase (Figure 4). The Michaelis–Menten curves for peroxidase-like activity towards H_2_O_2_ provided K_m_ and V_max_ values of 10.23 mM and 6.05 × 10^−8^ M s^−1^, respectively. Likewise, the oxidase-like activity was evaluated using TMB as substrate with respective K_m_ and V_max_ values of 0.123 mM and 2.38 × 10^−8^ M s^−1^ [53].

## 5. Nanozymes Involving Graphitic Carbon Nitride 

Graphitic carbon nitride (g-C_3_N_4_) is a metal-free 2D layered material composed of carbon and nitrogen, showing structural and functional resemblance to graphene and si- milar physical and chemical properties, although having superior biocompatibility, thermal stability, and solubility [61,62]. The nanosheets with stacked two dimensions, the sp2 hybridized N atoms in the tri-s-triazine and surface amino-groups, and the unique che- mical and physical properties have resulted in its widespread use in various sensing applications [63,64]. This carbon-based nanomaterial has been utilized to create highly active nanozymes for biosensing applications [65]. These mimicking enzymes have abundant pyridinic nitrogen moieties and a π-conjugated framework that provides potential bin-ding sites for further modifications to enhance their catalytic activity [66]. Indeed, g-C_3_N_4_ nanosheets have been shown to possess peroxidase-like activity, which has attracted attention not only in the field of sensors but also in the antibacterial area [67]. Furthermore, as seen for other carbon nanozymes, the catalytic performance of C_3_N_4_ nanosheets as artificial peroxidases has been improved by doping with other nanomaterials [13]. Several colorimetric applications based on the enzymatic activity of gC_3_N_4_ by TMB oxidation and color development have been reported [68,69]. However, few examples of electrochemical applications based on this activity have been found. Owing to its peroxidase-like activity, gC_3_N_4_ has been utilized as an artificial enzyme tag for signal amplification in electrochemical detection systems, the catalytic reaction leading to significant enhancement in voltammetric/amperometric signals [70]. Moreover, due to the strong electron-donating nature, this nanomaterial has a high electrocatalytic activity [71], so this property has been mainly exploited to anchor species and increase the selectivity in the design of electrochemical (bio)sensors. Table 4 summarizes the main characteristics of selected electrochemical (bio)sensors involving gC_3_N_4_ [72,73,74,75,76,77,78,79,80,81,82,83,84]. A hybrid nanozyme consisting of in situ growing PtNPs on gC_3_N_4_ nanosheets with enhanced peroxidase-mi-micking catalytic activity was prepared. Electrochemical experiments by cyclic voltammetry were performed to investigate the peroxidase-mimicking activity toward H_2_O_2_ [72]. Similarly, amperometry was used to study the performance and mechanism of peroxidase-like activity of oxygen-doped carbon nitride [73]. The electrocatalytic activity of gC_3_N_4_ with respect to its morphology was also investigated by studying the oxidation of ascorbic acid and dopamine and the reduction of hydrogen peroxide [85].

An electrochemical glucose sensor was prepared by Imram et al. [74] involving a gold electrode modified with ZnO/Pt/g-C_3_N_4_, where the enzyme-like activity of g-C_3_N_4_ was enhanced by doping with Pt and ZnO nanoparticles. These nanomaterials provided hydroxyl groups promoting the electrocatalysis in a neutral physiological buffer solution. The sensor exhibited a wide linear range from 0.25 to 110 mM of glucose and a good reproducibility in whole blood, which makes it suitable for diabetes monitoring. In other applications, the properties of g-C3N4 as an artificial enzyme and efficient transmitter of electrons allowed reinforcing the relatively low catalytic activity of the immobilized natural enzymes that arises from changes in the nature of the enzyme and the slow electron transfer process between the enzyme and the electrode [86]. An example is the method recently developed by Nasiri et al. [76] for the determination of lactose (Figure 5), in which the natural enzyme cellobiose dehydrogenase (CDH) was immobilized on chitosan-coated magnetic nanoparticles and the bioconjugate was anchored on gC_3_N_4_ sheets that acted as a direct electron transfer mediator to the electrode, thus increasing the electron transfer during the catalytic oxidation of lactose. The developed method allowed the determination of lactose in a 0.9 to 100 mM range with a detection limit of 0.3 mM and provided good results when applied to milk and dairy products.

Electrochemiluminescence (ECL) has increased the number of applications in the bio-sensing field due to its simplicity, low background, and high sensitivity [87]. Recent developments combining ECL with nanozyme amplification provide unique advantages for detection. For example, a cascade electrochemiluminescence (ECL) with integrated g–C_3_N_4_ @ MOF nanozyme was prepared for the immunosensing of PSA [77]. As Figure 6 shows, luminol was loaded inside the nanozyme by encapsulation of MOF (NH_2_-MIL (53)-Fe) within g–C_3_N_4_ nanosheets as a signal amplifier to label the detection antibody (Ab2). A sandwich-type immunosensing platform was developed with the capture antibody (Ab1) conjugated to magnetic nanoparticles (MNPs), providing ECL response in the presence of PSA. A calibration range of 0.1 pg mL^−1^ to 60 ng mL^−1^ and a high sensitivity were achieved with an LOD value of 0.02 pg mL^−1^. The good analytical performance was attributed to the high loading of luminol, the g-C_3_N_4_ @ MOF catalyzed cascade enzymatic mimetic reaction, and the highly efficient electron transfer at the nanoporous gold electrode (NPGE). This same group prepared an ECL immuno-DNA biosensor for methylated DNA in which the target was sandwiched between anti-5-methylcytosine monoclonal antibody conjugated to MNPs (MNPs/anti-5mc) and luminol loaded within a phosphorylated DNA capture probe immobilized onto g–C_3_N_4_ @ MOF (with MOF = UiO-66). In addition to the high signal amplification due to the nanozyme activity, this configuration showed a remarkable electrocatalytic activity of the rGO/pencil graphite electrode (PGE), providing a dynamic range from 20 pg to 20 ng with a detection limit of 10 pg [78].

Hybrids and nanocomposites of gC_3_N_4_ have also been used for the preparation of photoelectrochemical (bio)sensors [88] due to the need for the use of stable artificial enzymes [89] and the additional property of this nanomaterial as a photocatalyst that makes it possible to design schemes of signal amplification achieving higher sensitivity. As is known, in these sensors, the current response is generated by excitation from an external radiation source, and it is measured on a transparent electrode such as fluorine-doped tin oxide (FTO). As an illustrative example of these applications, Zhu et al. [81] prepared a nanozyme composed of gold nanoparticles and Cu^2+^-modified boron nitride nanosheets (AuNPs/Cu^2+^-B-g C_3_N_4_) for the construction of a signal-off aptasensor for the assay of telomerase activity. As Figure 7 shows, an FTO electrode modified with Ag_2_S/AgNP-decorated ZnIn_2_S_4_/C_3_N_4_ was used. Telomerase (TE) primer sequences (TS DNA) were extended by TE in the presence of deoxyribonucleoside triphosphates (dNTPs), which were bonded with the thiolated complementary DNA (cDNA). The nanozyme catalyzed the oxidation between 4-chloro-1-naphthol (4-CN) and H_2_O_2_ to generate insoluble precipitation on the photoelectrode, the inhibited signals with the TE-enabled TS extension allowing attaining a wide linear range of 50 to 5 × 10^5^ cells mL^−1^ and a low detection limit of 19 cells mL^−1^. 

## 6. Conclusions and Prospects

Carbon nanozymes exhibit several properties that make them attractive for their use in electrochemical (bio)sensing. Their stability, high conductivity, and surface properties, along with low toxicity and biocompatibility, are some of the most valued features for the design of biosensing platforms. In addition, these nanomaterials can be easily modified by incorporating functional groups or biomolecules (antibodies, aptamers, or even natural enzymes) to improve selectivity, as well as preparing hybrid or composite nanomaterials that provide greater sensitivity, usually due to the increased catalytic activity. The pro-perties of some carbon nanomaterials to act as artificial enzymes mimicking the role of natural peroxidases or oxidases provide an added value that makes them ideal for their use in environmental, food, or clinical fields, allowing the selective detection of target ana-lytes in complex samples. Apart from all these advantages, it should be noted that cu-rrently, the preparation of these nanomaterials is cost-effective since the synthesis procedures are mostly environmentally friendly and adhere to the principles of green chemistry, using raw materials of natural origin and with minimal reagent consumption and waste production. However, the future increase in the use of carbon nanozymes and the improvement in their performance in (bio)sensing and, particularly, in (bio)sensing using electrochemical detection requires solving some problems and facing some of the cha-llenges commented on below.

On the one hand, the preparation of carbon nanozymes with higher enzyme activity is an essential requirement. Indeed, in many cases, the artificial peroxidase or oxidase (the most common) role of these materials needs to be reinforced using less biocompatible and toxic metallic nanomaterials or even natural enzymes. In this sense, it is necessary to know the mechanisms by which the enzymatic activity is produced, since it can help to design more efficient nanomaterials. Furthermore, regarding electrochemical (bio)sensors, more research must be performed to clearly differentiate between electrocatalytic processes and those due to the enzymatic activity of the nanomaterial. In this sense, kinetic studies and verification of fitting to models such as the Michaelis–Menten type should be carried out. On the other hand, most of the developed methods involving carbon nanozymes use optical transduction based on the catalytic oxidation of TMB and colorimetric detection. However, despite the advantages of electrochemical detection, such as high sensitivity, low cost, portability, miniaturization capacity, and multiple analysis possibilities, the number of electrochemical biosensing methods using nanozymes is much smaller. An impulse should be given to these designs that involve the preparation of nanostructured electrode surfaces, making possible an efficient immobilization of reagents as well as developing strategies for signal amplification using nanozymes as labels or carrier tags, thus providing a more sensitive determination of analytes. Finally, the application of these (bio)sensors to real samples (clinical, environmental, or others) for the determination of the endogenous content of the target analytes in complex matrices is recommended to effectively validate the developed methods.

In summary, important issues and challenges still need to be solved to demonstrate the utility of carbon nanozymes in electroanalysis when facing real complex samples. These challenges should involve the development of new carbon nanomaterials with improved performance for analytical purposes as well as the development of methods of synthesis and preparation of composites and hybrids with enhanced enzyme activity able to allow the determination of endogenous species of interest at low concentration levels in complex real matrices.

## Figures and Tables

**Figure 1 micromachines-14-01746-f001:**
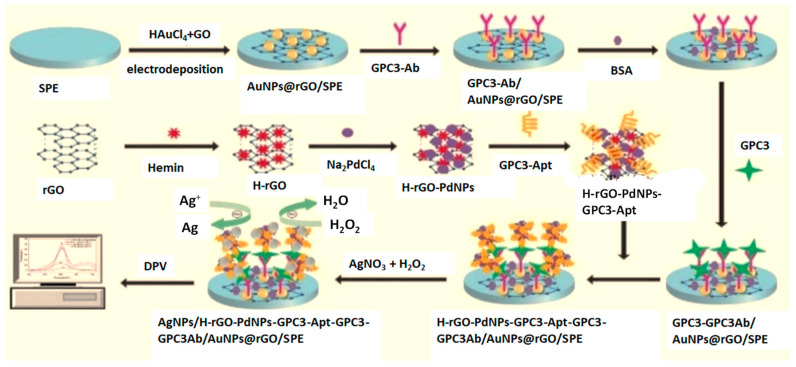
Schematic principle of the preparation of a GPC3 electrochemical nanobiosensor involving the H-rGO-Pd NPs nanozyme. Reproduced from [32] with permission.

**Figure 2 micromachines-14-01746-f002:**
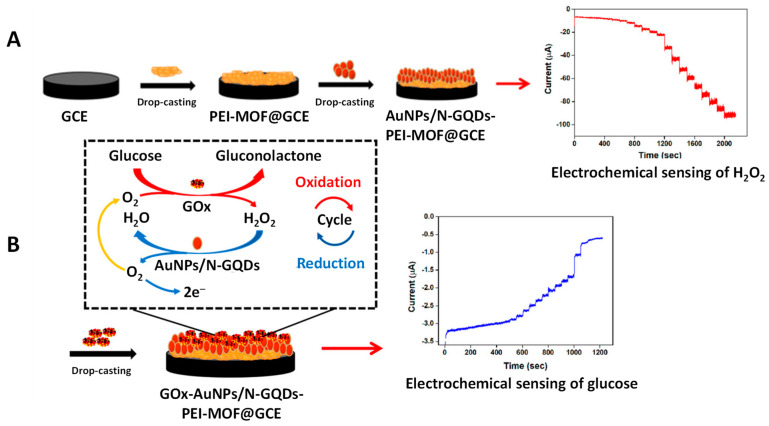
Illustration of the preparation procedure and sensing mechanism of AuNP/N-GQDs-PEI-MOF/GCE- and GOx/AuNP/N-GQDs-PEI-MOF/GCE-based amperometric sensors for (**A**) H_2_O_2_ and (**B**) glucose. Reproduced from [45] with permission.

**Figure 3 micromachines-14-01746-f003:**
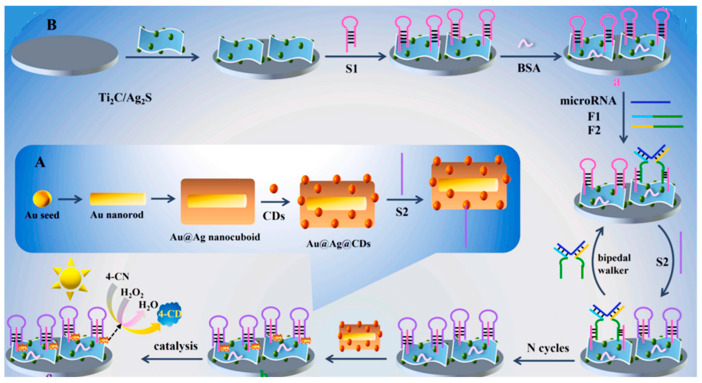
(**A**) Schematic display of the preparation of Ag@Ag@CDs; (**B**) Au@Ag@CDs nanoenzymes-driven multifunctional signal amplification combined with Ti_2_C/Ag_2_S composites for photoelectrochemical biosensing. Reprinted and adapted from [48] with permission.

**Figure 4 micromachines-14-01746-f004:**
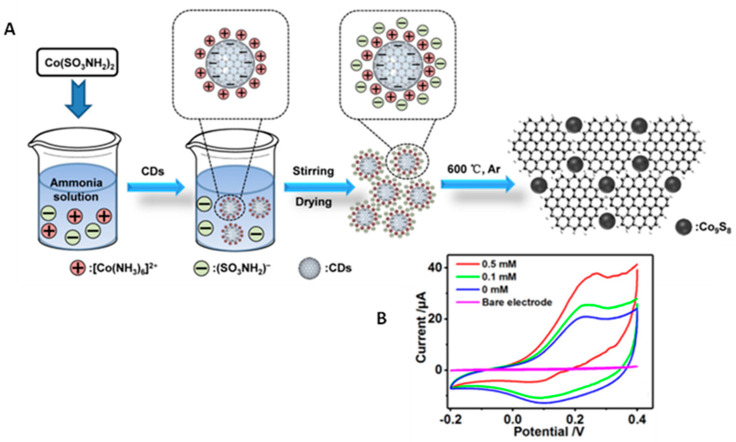
(**A**) Schematic illustration of the preparation of Co_9_S_8_/CDs and (**B**) cyclic voltammograms obtained at Co_9_S_8_/CDs/GCE with different concentrations of H_2_O_2_. Reprinted and adapted from [53] with permission.

**Figure 5 micromachines-14-01746-f005:**
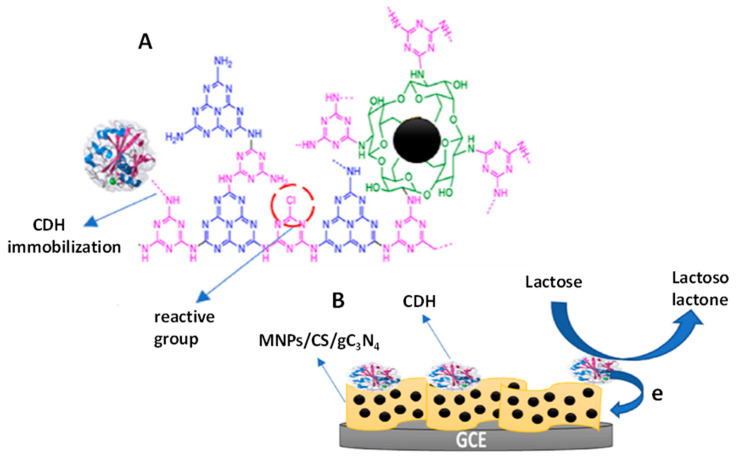
Schemes of (**A**) MNPs/CS/CDH/gC_3_N_4_ and (**B**) the electrotransfer mechanism during lactose oxidation. Reprinted and adapted from [76] with permission.

**Figure 6 micromachines-14-01746-f006:**
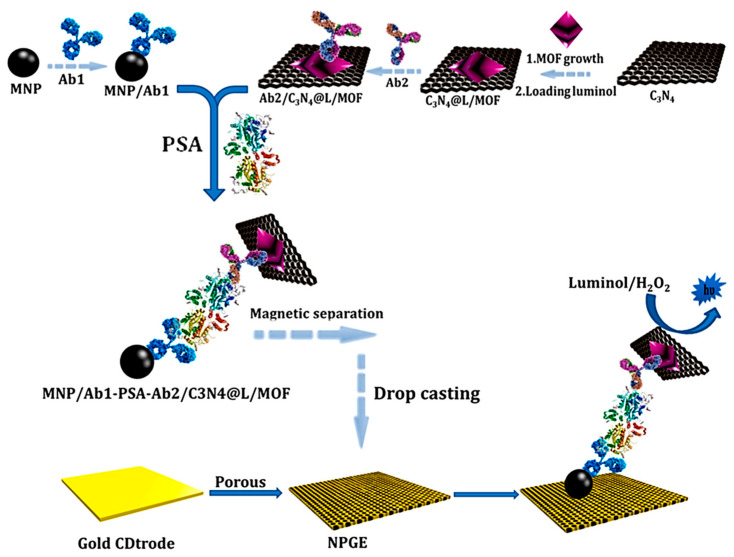
Amplified signal strategy based on the ECL-based nanozyme. Reprinted from [77] with permission.

**Figure 7 micromachines-14-01746-f007:**
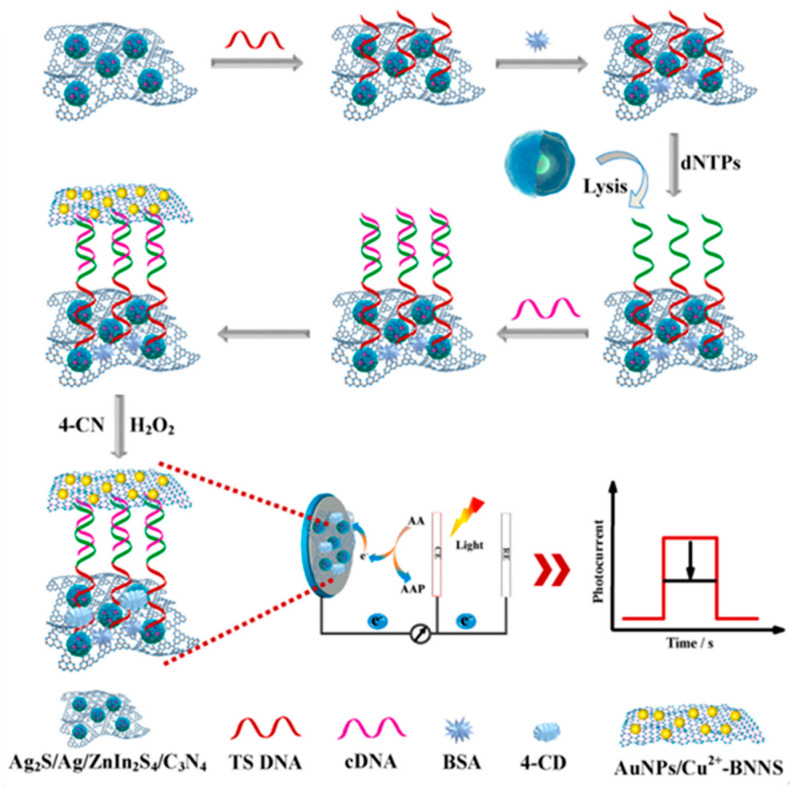
Schematic illustration of the photoelectrochemical biosensor developed for the determination of the telomerase activity. Reprinted from [81] with permission.

**Table 1 micromachines-14-01746-t001:** Some selected electrochemical (bio)sensors prepared with nanozymes involving carbon nanotubes.

Enzyme-like	K_m_/ Substrate	Configuration	Technique	Analyte	LR/LOD	Application/ Sample	Ref.
oxidase	87.48 μM/dopamine	Co/bNCNTs/GCE	DPV	dopamine	0.5–150 μM/ 0.0342 μM	Clinical/ spiked serum	[20]
oxidase	-	Fe/BNCNTs/GCE	amperometry	dopamine uric acid	1–630 μM/0.8 μM 0.5–2065 μM/0.28 μM	Clinical/ serum	[21]
oxidase	-	CNTs/ABTS/SPCE	CV	HSA	0.15–1.50 μM/ 60 nM	Clinical/urine	[23]
oxidase	-	MoS_2_/MWCNTs/ GCE	DPV	CBZ	0.04–100 μM/ 7.4 nM	Food/tea and rice	[24]
peroxidase	-	MWCNTs/MoS_2_ (-HRP)-dAb (nanocarrier)	amperometry	BAFF/APRIL	0.24–120 ng/mL/ 0.08 ng/mL (BAFF); 0.19–25 ng/mL/ 0.06 ng/mL (APRIL)	Clinical/ serum and cell extracts	[26]

Abbreviation: ABTS, 2,20-azino-bis(3-athylbenzothiazoline-6-sulfonic acid); APRIL, a proliferation-inducing ligand; BAFF, B cell activating factor; BNCNTs, boron–nitrogen-co-doped carbon nanotubes; bNCNTs, bamboo-like N-doped carbon nanotubes; CBZ, carbendazim; CV, cyclic voltammetry; dAb, detection antibody; DPV, differential pulse voltammetry; LR, linear range; and LOD, limit of detection.

**Table 2 micromachines-14-01746-t002:** Some selected electrochemical (bio)sensors prepared with graphene-based nanozymes.

Enzyme-like	K_m_/ Substrate	Configuration	Technique	Analyte	LR/LOD	Application/ Sample	Ref.
oxidase	-	GO-COOAuNPs/GCE	amperometry	glucose	0.02–4.48 mM/6 μM	Clinical/serum	[31]
peroxidase	-	H-rGO-PdNPs-GPC3Apt/GPC3/GPC3Ab	DPV	glypican-3	0.01–10.0 μg/mL/3.30 ng/mL	Clinical/spiked serum	[32]
oxidase	35.38 μM/ dopamine 51.17 μM/UA	CoO/NCS-rGO/GCE	DPV	dopamine uric acid	0.5−110 μM/0.15 μM 1−125 μM/0.22 μM	Clinical/spiked serum	[33]
laccase	1.87 mM/catechol	AuNPs/rGO/SPE	DPV	catechol	1.0 nM–1.0 mM/0.33 nM	Environmental /spiked waters	[34]
oxidase	-	CuNCs/graphene/SPCE	DPV	dopamine	0.001–100 μM/0.33 nM	Clinical/plasma	[35]
superoxide peroxidase catalase	9.2 mM/TMB	PBNPs/rGO/GFE	CV	H_2_O_2_	0.0012–5 mM/1.2 μM	-	[36]
oxidase	72.49 μM/X 66.84 μM/HX	PI/graphene flexible sensor	DPV	xanthine (X); hypoxanthine (HX)	0.3–179.9 μM/0.26 μM; 0.3–159.9 μM/0.18 μM	Food/fish	[37]
peroxidase	-	rGO-CMC- hemin@Pt/AuNPs/SPCE	DPV	1,5-anhidro- glucitol	0.1–2.0 mg/mL/38.2 μg/mL	Clinical/spiked serum	[38]
oxidase peroxidase catalase	0.57 mM/H_2_O_2_ 0.074 mM/TMB	PdNPs/ N-PC/rGO/GCE	amperometry	glutathione	70 nM–1500 μM/9.8 nM	Clinical/serum	[39]
oxidase	-	Fe_3_O_4_/MGO/ITO	amperometry	glucose	0.1−16 mM	-	[40]

Abbreviation: CMC, carboxymethylated chitosan; CuNCs, copper nanocubes; CV, cyclic voltammetry; DPV, differential pulse voltammetry; GPC3, glypican-3; ITO, indium tin oxide electrode; LR, linear range; LOD, limit of detection; NCS, nitrogen-doped carbon sheets; PB, Prussian Blue; PC, porous carbon; PI, polyimide; rGO, reduced graphene oxide; and TMB, 3,5,3,5-tetramethylbenzidine.

**Table 3 micromachines-14-01746-t003:** Some selected electrochemical (bio)sensors involving graphene quantum dots and carbon dots nanozymes.

Enzyme-like	K_m_ (mM)/ Substrate	Configuration	Technique	Analyte	LR/LOD	Application/ Sample	Ref.
oxidase	0.11/TMB	MnO_2_/GQD/SPE	DPV	DA, catechol	0.5–100 μM/0.05 μM 5–150 μM/0.09 μM	Environmental/waters	[42]
oxidase	-	AuNPs/GQDs/SPCE	SWV	quercetin	0.1 nM–1 mM/ 0.033 nM	Clinical/plasma	[43]
-	-	ANSA/AuNPs/GQD/GCE	DPV	MTX	0.1–100 μM/ 0.03 μM	Clinical/plasma	[44]
peroxidase	0.056/TMB 0.469/H_2_O_2_	AuNPs/N-GQDs-PEI-MOF/GCE; GOx/AuNPs/N-GQDs-PEI-MOF/GCE	amperometry	H_2_O_2_ glucose	5 μM–1 mM; 2–10 mM/3.38 μM (H_2_O_2_); 2–10 μM; 0.02–3 mM; 0.7 μM (glucose)	Clinical/serum	[45]
undefined	-	anti-cTnI-AuNPs @GQDs/SPGE	SWV	cTnI	1–1000 pg mL^−1^ /0.1 pg mL^−1^	Clinical/serum	[46]
peroxidase	0.0196/TMB 34.76/H_2_O_2_	AuNPs/N-CDs/GCE	CV; amperometry	H_2_O_2_	-	-	[47]
peroxidase	-	B-CDs-AuNPs/AuE	amperometry	ompA; *C. sakazakii*	0.001–10 pM/ 0.04 fM (ompA) 7.8–7.8 × 10^6^ CFU mL^−1^ 2.6 CFUmL^−1^	Food/infant formula	[48]
peroxidase	-	Ti_2_CMxenes/Ag_2_S CDs/ITO	photoelectrochemistry	microRNA-155	1–10,000 fM /0.83 fM	Clinical/serum	[49]
peroxidase	-	MIP/Fe_3_O_4_/CDs @Ag-MOF/AuE paper-based microfluidic	DPV	parathion-methyl	0.05–20 nM/ 0.0116 nM	Food, environmental/foods, soil, and water	[50]
peroxidase	-	PDDA/CuO/CDs/ SPCE	amperometry	glucose	0.5–2 mM; 2–5 mM/0.2 mM	Clinical/spiked serum	[51]
peroxidase	67.2/H_2_O_2_	Fe_3_O_4_/CeO_2_/CDs/ MWCNTs/ILs	CV	H_2_O_2_	0.02–1.0 μM/0.02 μM	-	[52]
peroxidase oxidase	0.64/TMB 10.23/H_2_O_2_	Co_9_S_8_/CDs/GCE	CV	H_2_O_2_	0.1; 0.5 mM	-	[53]

Abbreviations: ANSA, 1-amino-2-naphthol-4-sulfonic acid; B-CDs, boron-doped carbon dots; CV, cyclic voltammetry; ILs. ionic li-quids; MIP, molecularly imprinted polymer; MOF, metal organic frameworks; MTX, methotrexate; PDDA, polydiallyl-dimethyl-ammonium; PEI, polyethyleneimine; and SWV, square-wave voltammetry.

**Table 4 micromachines-14-01746-t004:** Some selected electrochemical biosensors involving graphitic carbon nitride (gC_3_N_4_) nanozymes.

Enzyme-like	K_m_ (mM) Substrate	Configuration	Technique	Analyte	LR/LOD	Application/ Sample	Ref.
oxidase	0.105/H_2_O_2_ 0.446/TMB	PtNPs@gC_3_N_4_/ GCE	CV	H_2_O_2_	-	-	[72]
peroxidase	0.098/TMB 0.44/H_2_O_2_	O-gC_3_N_4_	amperometry	H_2_O_2_	-	-	[73]
peroxidase	116/glucose	ZnO/Pt/gC_3_N_4_/ AuE	amperometry	glucose	0.25–110 mM/0.1 mM	Clinical/serum urine blood	[74]
-	-	MB/Au@Pt-Apt-SMZ-Ab1-AuOct/PEI/cC_3_N_4_/AuE	SWV	SMZ	0.1 pg mL^−1^–100 ng mL^−1^ 0.069 pg mL^−1^	Food/milk	[75]
-	-	MNPs/CS/gC_3_N_4_/CDH/GCE	amperometry	lactose	0.9–100 mM/0.3 mM	Food/milk products	[76]
peroxidase	0.022/TMB 0.136/H_2_O_2_	MNPs/cAb1-PSA-Ab2/C_3_N_4_ @LUM/NMF/NPGE	ECL	PSA	10^−4^–60 ng mL^−1^ /0.02 pg mL^−1^	Clinical/spiked plasma	[77]
peroxidase	-	MNP/anti-5mc rGO/PGE DNA	ECL	methylated DNA	20 pg–20 ng/10 pg	Clinical/plasma	[78]
peroxidase	-	gC_3_N_4_/BN/GCE	amperometry	HQ	0.02–0.08 μM; 0.09–0.17 μM/0.009 μM	Environmental/waters	[79]
peroxidase	-	Ag/gC_3_N_4_/GCE	DPV	HQ	0.99–999.96 μM/5.8 μM	Environmental/waters	[80]
peroxidase	-	Ag_2_S/Ag/ZnIn_2_S_4_/C_3_N_4_/FTO	photo- electrochemistry	telomerase activity	50–5 × 10^5^ cells mL^−1^/ 19 cells mL^−1^	Clinical/HeLa cells	[81]
peroxidase	-	CoOOH/gC_3_N_4_/ CuInS_2_/FTO	photo- electrochemistry	CEA	0.02–40 ng mL^−1^/ 5.2 pg mL^−1^	Clinical/serum	[82]
-	-	MIP/GQDs/ B-gC_3_N_4_/GCE	DPV	BPA	0.01–1 nM/3 pM	Food/orange juice	[83]
-	-	anti-AFB1/Thi/gC_3_N_4_/ ITO	CV	AFB1	1 fg mL^−1^–1 ng mL^−1^/ 0.328 fg mL^−1^	-	[84]

Abbreviations: Ab1, capture antibody; Ab2, detection antibody; AFB1, aflatoxin B1; cC_3_N_4_, carboxylated carbon nitride; CDH, cellobiose dehydrogenase; CEA, carcinoembryonic antigen; HQ, hydroquinone; LUM, luminol; MNPs, magnetic nanoparticles; NMF, NH_2_-MIL(53)-Fe; PEI, poly (ethylenimine); PGE, pencil graphite electrode; PSA, prostate-specific antigen; SMZ, sulfamethazine; SWV, square wave voltammetry; and Thi, thionine.

## Data Availability

Not applicable.

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
