# Peer review of "Carbon-Based Enzyme Mimetics for Electrochemical Biosensing"

_micromachines, 2023, doi:10.3390/mi14091746_

Round 1

Reviewer 1 Report

Authors have presented a comprehensive study of carbon-based enzymatic biosensors, however my comments are below to revise the content to improve the quality.

1. Include newer carbon materials in the review.

2. Each carbon material section should be sub categorized based on the application for example, agriculture, environmental monitoring, biomedical applications etc.

3. Apart from the basic performance of biosensors (LOD) and real sample analysis, application should be included in the table.

4. CNTs and graphene should be described separately.

Quality of english should be revised for  minor grammar corrections and long phrases should be rewritten

Author Response

Reviewer 1.

Authors have presented a comprehensive study of carbon-based enzymatic biosensors, however my comments are below to review the content to improve the quality.

Thank you very much for the Reviewer's comment and the useful recommendations.

  1. Include newer carbon materials in the review.

Thank you for the comment. In fact, we have included in the article the different carbon nanomaterials used as nanozymes for the preparation of electrochemical (bio)sensors due to their catalytic properties. Of course, we have reviewed the carbon nanomaterials reported in the literature during the last four years which include the newer materials such graphitic carbon nitride.

  1. Each carbon material section should be subcategorized based on the application for example, agriculture, environmental monitoring, biomedical applications etc.

Unfortunately, the number of applications in each category is not enough -in some cases there are none- to organize subsections devoted to different analytical fields. However, we have now included in the Tables the field of application of the different designs. Moreover, some sentences highlighting the main analytical fields of carbon nanozymes have been included in the Conclusions section.

  1. Apart from the basic performance of biosensors (LOD) and real sample analysis, application should be included in the table.

Thank you for this comment. Accordingly, we have added to the tables the application data provided by the authors of the corresponding articles.

  1. CNTs and graphene should be described separately.

Thank you very much for your suggestion. We have discussed separately the two nanomaterials in the revised manuscript and summarized the examples in individual tables (Tables 2 and 3).

Quality of English should be revised for minor grammar corrections and long phrases should be rewritten

Thanks again. We have rewritten the article reviewing in depth the English wording and grammar.

Author Response

Reviewer 2.

In this review article the authors focused on the use of carbon-based nanozymes for the preparation of electrochemical (bio)sensors. This is an interesting review in perspective of enzymatic biosensors development. I have few minor comments before accepting it for publication.

We appreciate your feedback and helpful recommendations.

  1. Figure 2 needs to be replotted. The image resolution is too low.

Thank you very much. We have included a new Figure 2 in the revised manuscript.

  1. This review needs to provide a separate section on “Challenges and issues for nanozymes”.

Thank you very much for your recommendation. We think that a separate section "Challenges and issues for nanozymes" would be appropriate in an article dedicated to nanozymes in general. However, considering that the article is focussed only in carbon-based nanozymes, we have preferred to include an additional paragraph in the Introduction section, highlighting even more the advantages and general applications of these nanozymes. Furthermore, according to the Reviewer´s comment, we have added in the Conclusions and Future Prospects section, a broad paragraph on Challenges and Issues for carbon nanozymes.

  1. There are few typing errors in superscript and subscript (e.g., O2Ë™-) that need to be corrected.

Thank you very much. We have revised the manuscript and the typographical errors have been corrected in the new version.

Round 2

Reviewer 1 Report

The authors addressed the issues raised and revised the manuscript thoroughly. The manuscript can be accepted in the present form after minor English and grammar correction.

Minor English and grammar correction is needed.